# SmartX Box: Virtualized Hyper-Converged Resources for Building an Affordable Playground

**Aris Cahyadi Risdianto**, **Muhammad Usman** and **JongWon Kim** *

School of Electrical Engineering and Computer Science, Gwangju Institute of Science and Technology (GIST), Gwangju 61005, Korea; aris@nm.gist.ac.kr (A.C.R.); usman@smartx.kr (M.U.)

*   Correspondence: jongwon@nm.gist.ac.kr; Tel.: +82-62-715-2219

**Abstract:** In this paper, we present our proposals and efforts for building an affordable playground (i.e., miniaturized testbed) for Software-Defined Networking (SDN)-Cloud experiments by using hyper-converged SmartX Boxes that are distributed across multiple sites. Each SmartX Box consists of several virtualized functions that are categorized into SDN and cloud functions. Multiple SmartX Boxes are deployed and inter-connected through SDN to build multi-site distributed cloud playground resources. The resulting deployment integrates both cloud multi-tenancy and SDN-based slicing, which allow developers to run experiments and operators to monitor resources in a distributed SDN-cloud playground. It also describes how the hyper-converged SmartX Box can increase the affordability of the playground deployment. Thus, the analysis result shows the efficiency of SmartX Box for building a distributed playground by providing semi-automated DevOps-style resource provisioning.

**Keywords:** affordable playground; hyper-converged SmartX Box; distributed resources; multi-site and virtualized cloud; software-defined networking; DevOps automation

---

## 1. Introduction

### 1.1. Background

Aligned with worldwide Future Internet testbed efforts (e.g., GENI—Global Environment for Network Innovations [1], FIRE—Future Internet Research and Experimentation [2]), OF@TEIN (OpenFlow at Trans-Eurasia Information Network) project was started to build an OpenFlow-enabled testbed over TEIN infrastructure in 2012 [3]. Several experimentation tools were developed to support both developers and operators in using OF@TEIN testbed. Initially, a mixed combination of tools, ranging from simple web-/script-based to DevOps (Development and Operations) [4] Chef-based automated tools [5], were deployed over SmartX Racks. SmartX Rack consists of four devices: *Management & Worker node, Capsulator node, OpenFlow switch,* and *Remote power device*. Physically LAN-connected SmartX Racks were inter-connected by L2 (layer 2) tunnels, employed in Capsulator nodes as described in detail in this work [6]. However, SmartX Racks with multiple devices were subject to physical remote re-configurations, which are extremely hard to manage for distributed OF@TEIN Playground. Thus, from late 2013, a hyper-converged SmartX Box was introduced to virtualize and merge the functionalities of four devices into a single box [7]. The comparison between previously deployed SmartX Racks and newly deployed SmartX Box, is depicted in Figure 1. Finally, several hyper-converged SmartX Boxes were distributed deployed over nine Asian countries in 2015, as shown in Figure 2.

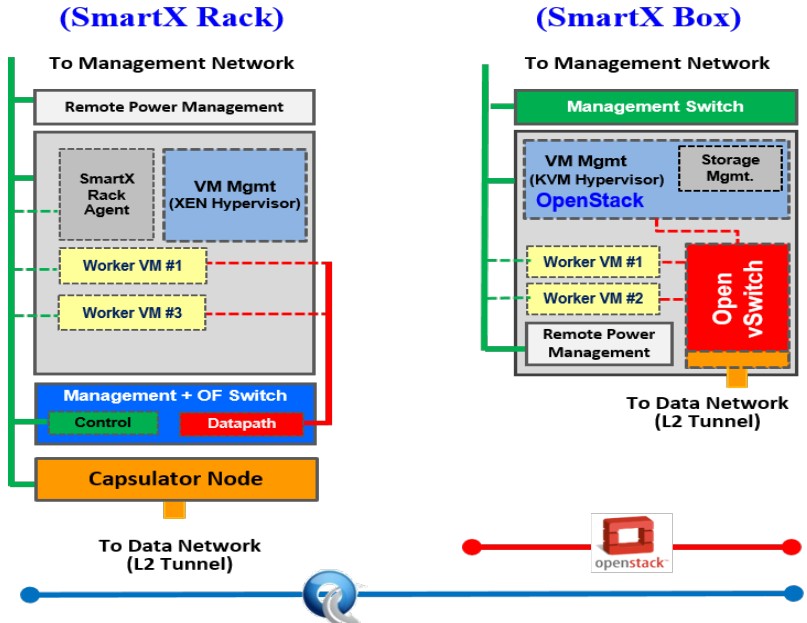

**Figure 1.** SmartX Rack versus SmartX Box Comparison.

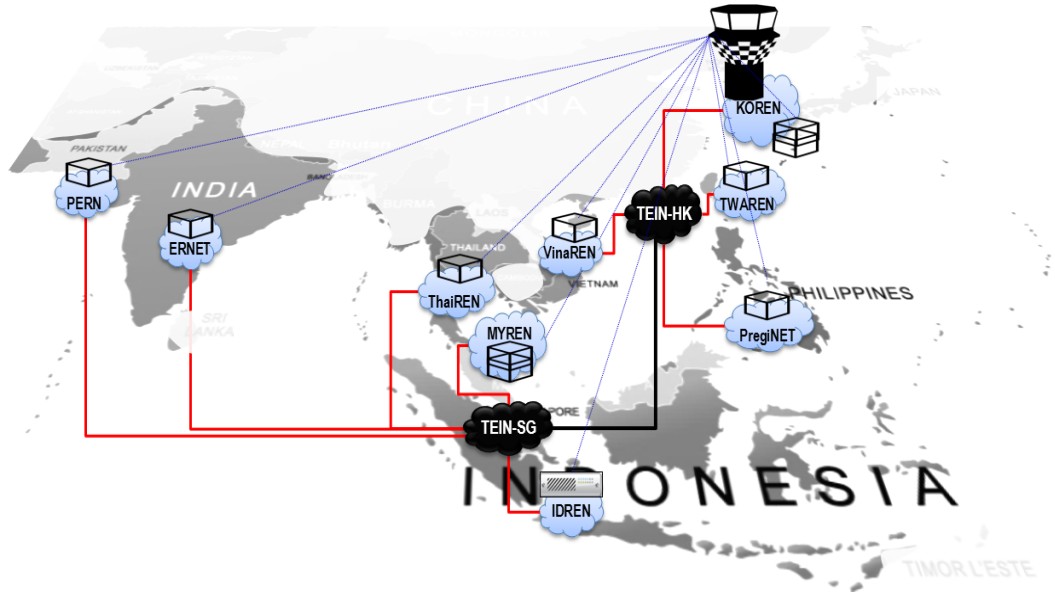

**Figure 2.** OF@TEIN Playground.

Software-defined Networking (SDN) tools assist developers and operators to prepare the experiment environment in OF@TEIN, by enabling networking resources (e.g., switches and Flowspaces [8]) preparation. Similarly, cloud management software can cover computing resources (e.g., VMs) preparation. Thus, the combination of distributed SDN and cloud testbeds ready to provide scalable and flexible computing resources with enhanced networking capability. However, the seamless integration of SDN-enabled and cloud-leveraged infrastructure is a very challenging task, due to open and conflicting options in configuring and customizing resource pools together. Therefore, we should carefully provision all resource configuration aspects such as multi-site distribution, virtualized resource slicing (i.e., isolation), and multi-tenancy support while considering hardware deployment for SDN and cloud integrated testbed.

As mentioned above, this paper proposes the concept of hyper-converged SmartX Box that can easily accommodate virtualized and programmable resources (i.e., OpenFlow-enabled virtual switches and OpenStack-leveraged cloud VMs) to build the OF@TEIN SDN-enabled multi-site clouds playground (i.e., miniaturized testbed). The collection of OpenFlow-based virtual switches (i.e., OpenvSwitch [9]) is providing SDN capability, which is controlled by both developers and operators SDN controllers. Simultaneously, OpenStack-leveraged VMs (i.e., working as virtual Boxes) are effectively managed by OpenStack cloud management [10]. However, the actual design and implementation of SmartX Box are continuous to evolve for supporting new experiment over OF@TEIN Playground.

### 1.2. Motivation and Related Work

It is well known that the service-centric networking model to provide higher-level connectivity and policy abstraction is an integral part of cloud-leveraged applications. The emerging SDN paradigm can provide new opportunities to integrate cloud-leveraged services with enhanced networking capability through deeply programmable interfaces and DevOps-style automation. Several SDN-based approaches have been proposed to provide virtualized overlay networking for multi-tenancy cloud infrastructure. For example, Meridian proposed an SDN controller platform to support service-level networking for cloud infrastructure [11]. Similarly, CNG (Cloud Networking Gateway) attempts to address multi-tenancy networking for distributed cloud resources from multiple providers while providing flexibilities in deploying, configuring and instantiating cloud networking services [12]. As a prototype of implementation, the large-scale deployment of GENI Racks over national R&E (research and education) network is also moving towards a programmable, virtualized, and distributed collection of networking/compute/storage resources, a global-scale "deeply programmable cloud". It satisfied research requirements in a wide variety of areas, including cloud-based applications [1]. Another effort from the EU, known as "BonFIRE", is a multi-site testbed that supports testing of cloud-based distributed applications, which offer a unique ease-to-use functionality in terms of configuration, visibility, and control of advanced cloud features for experimentation [13].

Aligned with converged SDI (Software-defined Infrastructure) paradigm [14], the SDN and cloud testbeds should continuously support for the new type of technologies and experiments. To guarantee the usability and continuity of the testbed, the testbed deployment needs to consider the three following aspects. First, the resources should be open without any limitations due to proprietary vendor software or hardware implementation. Second, the resources should be conceptually and reality agile to adopt new technologies or match experiment requirements. Third, the cost and number of physical (i.e., hardware) resources to be deployed should be very minimum for each site of the playground. It only requires a minimized budget (i.e., leveraging low-cost commodity hardware). Please note that leveraging the power of open-source software (e.g., KVM hypervisor [15], LXC Linux container [16], Open virtual switch [9], and others) is also important to support distributed resources centralized management.

Since the cloud computing model is massively adopted for computation infrastructure deployment, it should consider an affordability aspect of the cloud deployment model. In 2017, Ta A.D. [17] tried to address the adoption of a cloud computing framework for developing countries by leveraging hyper-converged servers as virtual computing infrastructure. Moreover, it has gained more acceptance since the networking function (i.e., switch) was extended into a virtualization layer, which allows the creation of multiple virtual switches in the Linux-based server [9]. Then, in 2016, a software-defined framework, called *OpenBox*, tries to address NFs (network functions) deployment by decoupling between NFs control-plane and data-plane that is very similar to SDN solutions [18].

### 1.3. Aim and Contributions of the Paper

We tried to align with the above testbed deployment, but unfortunately, there are some technical and environment gaps with all those testbed efforts. GENI [1] covers extensive testbed features and functionalities for a different type of experiments while considering increasing the performance with powerful hardware. However, they are not focusing on distributed over heterogeneous infrastructure and the cost of deployment due to the good support of infrastructure and funding sources. FIRE [2] is similar to GENI, but with a less extensible feature because they have much more focus on smaller testbed but also with good performance. They are a little bit concerned about the distributed over heterogeneous infrastructure due to different conditions and policies among many Europeans countries. PlanetLab [19] does not consider features and performance due to the simple requirement of virtual resources with basic networking. However, it is a large-scale distributed testbed because it is deployed over a thousand sites with different network infrastructures. PlanetLab does not concern about the deployment cost because it is supported by a large research community in adding new hardware resources as a new testbed site.

In summary, it is not possible to directly apply their deployment approaches into our environment due to several reasons such as limited funding, level of research interest, and network infrastructure support. Therefore, we want to provide extensive experiment features/functions over a reasonable number of distributed sites over heterogeneous infrastructure. However, at the same time, we need to reduce the number of hardware for each experiment by providing hyper-converged box-style resources. In other words, we want to increase the testbed capability while keeping the cost of the deployment as low as possible. Thus, since 2013, OF@TEIN had considered those mentioned aspects by changing the initial rack-style resource deployment from GENI testbed into box-style, hyper-converged, and server-based resources for an affordable playground across developing countries in Asia region [7]. The main contributions of this paper are:

1. Proposing a concept of an affordable playground with a centralized playground tower and multiple centers to manage and control distributed resources spread across heterogeneous infrastructure.
2. Proposing a design of box-style resources (i.e., SmartX Box), an *open, agile* and *economic* hyper-converged resources which able to be deployed and verified over underlying heterogeneous infrastructure.
3. Reducing the cost of the deployment and provisioning time of the playground by leveraging low-cost commodity hardware and developing the open-source-based DevOps automated tools to provision hyper-converged box-style resources.

## 2. An Affordable Playground

### 2.1. An SDN-Enabled Multi-Site Clouds Playground

A *Playground* is defined as a miniaturized and customizable testbed that is easy to build and operate for various research experiments by a tiny-size DevOps-style team of developers and operators. Mainly, we focus on establishing a multi-site playground infrastructure where playground resources are physically distributed across multiple geographical sites but logically inter-connected with each other to offer a unified shared pool of resources. A tiny-size team of people (i.e., operators) should provision and control the multi-site isolated resources, which is openly accessible by a group of people (i.e., developers) from all involved sites. As depicted in Figure 3, the proposed multi-site playground has several vital entities, which are discussed below:

1. **Playground Sites with Hyper-converged Boxes**. When playground developers want to perform their experiments, they can dynamically acquire dedicated resources from the pool of multi-site resources. For the customizable (i.e., software-defined) playground, the resource infrastructure of the multi-site playground should be composable (e.g., programmable) in terms of computing, storage, and networking types of resources. By leveraging the growing popularity toward

hyper-converged appliances that integrate computing, storage, and networking resources into a single box-style entity, we can enable the multi-site resource pool that ready to be customized and scaled-out without the manual intervention of playground operators. Thus, in our approach, each playground site is equipped with a box-style of hyper-converged resources, denoted as *SmartX Box*, the hyper-converged box-style resource that should be useful in supporting the required composability by comfortably accommodating virtualized and programmable resources. Multiple types of SmartX Box should be designed and deployed for different purposes, such as SD-WAN, SDN-enabled clouds, and access extension support. However, the critical aspects of those boxes design are similar: *open, agile*, and *economic* resources.

Furthermore, the SmartX Box can be divided into three main abstractions, which are *box, function,* and *inter-connect*. The *box* represents all the server-based hardware that runs Linux as a baseline open-source operating system for other software-based functions. The *function* represents virtual functions (e.g., virtual machine, virtual switch, or virtual router), which are implemented by using a set of open-source software. Finally, the *inter-connect* is representing the path/link between functions or boxes which include the tunnel-based overlay networking because playground sites are spread over the heterogeneous underlay network infrastructure.

2. **Playground Tower with Centers**. To satisfy the dynamic requirements of playground developers on diversified functionalities over distributed but miniaturized resource pools, the proposed playground should integrate the emerging technology paradigms such as the SDN, cloud computing, and Internet of Things (IoT). However, the SDN-enabled multi-site clouds combination brings new complexities for the playground operators, since multi-site clouds resources, connected via SDN-based networks, demand various software-based DevOps-automation tools to build, operate, and use automatically. Thus, we propose the concept of *Playground Tower*, which provides a logical space-like abstraction in a centralized location, which leads the operation of the multi-site playground by following the concept of "monitor and control" tower. From the tower, the DevOps team can enjoy a panoramic view of playground resources that are distributed over underlay networks, and quickly manage and use those resources for their experiments.

   The playground tower systematically covers various functional requirements of operating a multi-site playground by employing several centers, also depicted in Figure 3. First, Provisioning Center (P-center) is responsible for remote installation and configuration of multi-site playground resources. Visibility Center (V-center) covers playground visibility and provides panoramic visualization support. Orchestration Center (O-center) handles the management level issues with the assistance of controllers (e.g., SDN/cloud controllers). Thus, several software-based tools are used and developed by leveraging open-source software. For example, to provision SmartX Box, P-Center provides an automation framework such as Chef and MaaS. To continuously operate by re-configuring the playground resources, O-Center provides a set of interfaces (e.g., CLI, API, Web UI) to meet the varying requirements of playground developers. Finally, to monitor the playground resources and traffic flows, V-Center provides playground visibility data in an accessible format for visualization and analysis.

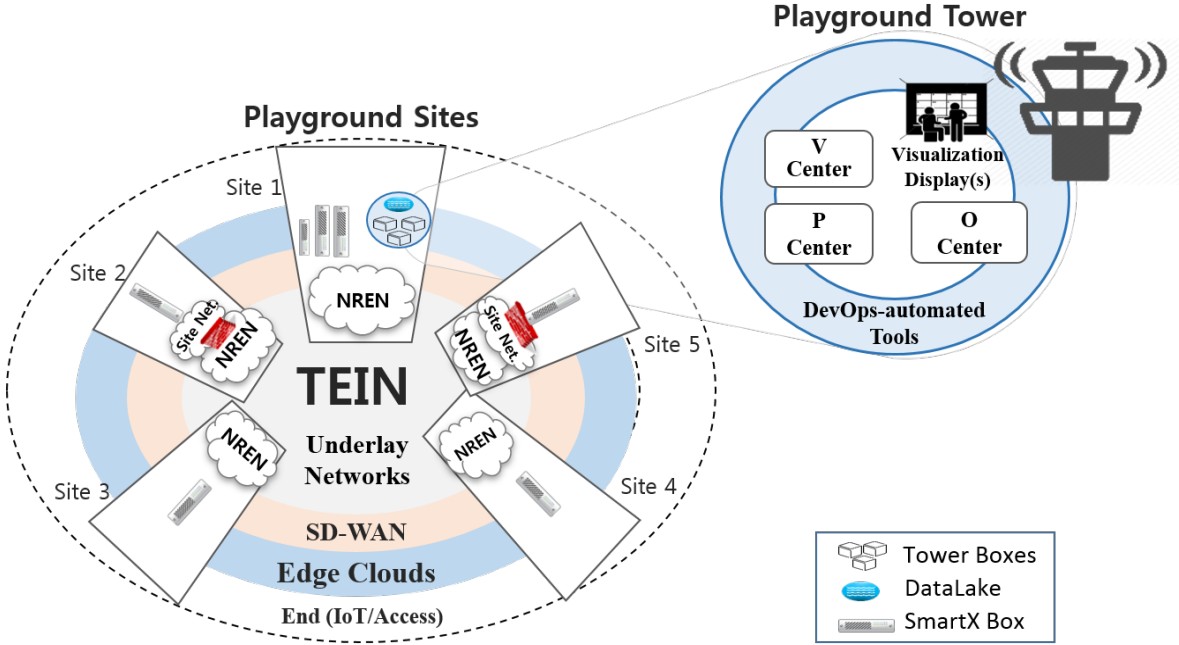

**Figure 3.** An SDN-Enabled Multi-Site Clouds Playground.

### 2.2. Affordable Playground with SmartX Box

As mentioned earlier, to specifically address the affordability of multi-site playground, the resources should be open without any configuration/monitoring limitations, agile to adopt new technologies or match experiment requirements, and economics by the deployment of low-cost commodity hardware. Open-source software in converged SDI with SDN, cloud, and Network Function Virtualization (NFV) integration, and the support of open-source hardware are the main drivers of the development of hyper-converged box-style resources.

The concept of box-style hyper-converged resources is shown in Figure 4, which can introduce a unique design of an affordable multi-site playground. The critical design of hyper-converged box-style resource is open, agile, and economic resources. The term box is adopted from a white box (i.e., clear box or open box) that has understandable and controllable subsystems, so it is easy to develop/test software on it without limitation on a vendor-specific feature. The box is open for supporting any server-based hardware because it is fundamentally characterized by software-based components/functions composition and implementation. It is agile to provide a different type of experiment for matching with new upcoming technologies and requirements. It is economical to allow any low-cost commodity server to be used to increase affordability in distributed deployment and operation. However, the actual design of the box is evolved, which include: simplification of the physical hardware specification, components design changes, and modular software implementation.

Unfortunately, provisioning and operating distributed hyper-converged boxes in heterogeneous physical infrastructures quite challenging since it is subjected to different performance parameters (e.g., networks speed and power stability). Also, the independency between multi-domain underlying infrastructure operators (e.g., access and security policies). Consequently, it is hard to maintain the continuous operation of all the boxes. To make those boxes design turnkey simple, we should concentrate on minimizing the requirement and consumed time for provisioning the boxes with these following strategies:

1. *Heterogeneous hardware*: No specific hardware requirements or specific brand/vendor supports, all hardware with acceleration support for virtualization, and multiple network interfaces support for specialized connections can be used.

2.  *Remote management*: A distributed deployment and limited access to installation sites are the main reasons to enable the hardware remote management module based on an Intelligent Platform Management Interface (IPMI) [20].
3.  *Automated provisioning*: A set of DevOps-based automated provisioning tools is developed to minimize the consumed time for installing and configuring the box with pre-arranged multiple specialized connections.
4.  *Centralized configuration template*: The provisioning tool is equipped with a specific template of the configuration file for a specific site of the box that downloadable from a centralized location with a specific component configuration for SDN and cloud.

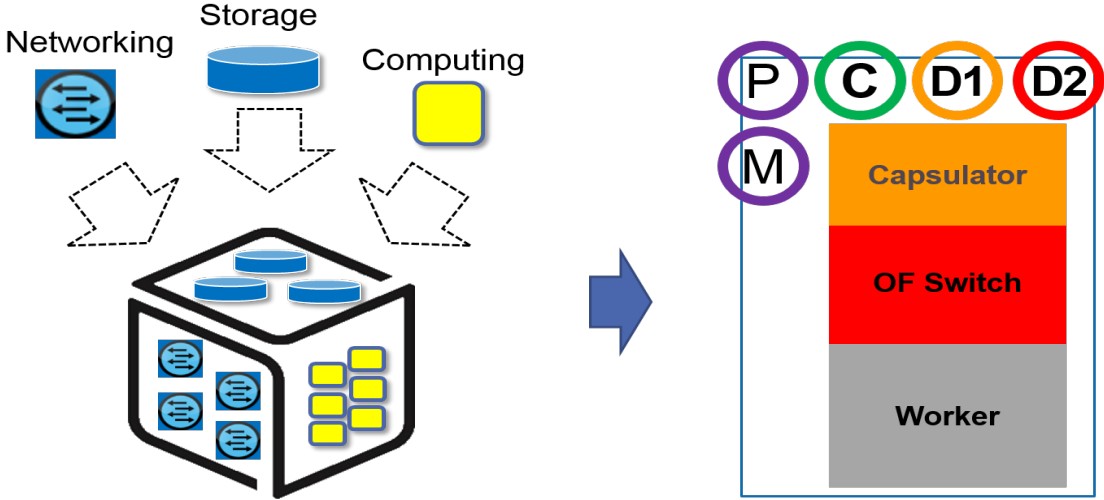

**Figure 4.** The concept of box-style hyper-converged resources.

*2.3. SmartX Box: Design and Implementation*

2.3.1. SmartX Box Abstraction

As discussed above, we adopt the hyper-converged box-style resource, called *SmartX Box* [7], as the main building block for our OF@TEIN Playground. The proposed SmartX Box illustrated in Figure 5, as an abstracted format.

The SmartX Box abstraction tries to align the SDN/NFV/cloud integration of SDI with the compute/storage/networking integration of hyper-converged box-style resources. Cloud can provide economics and scalable computing/storage resources without compromising the associated performance, availability, and reliability. SDN provides flexible networking support for highly virtualized computing/storage resources, which is not possible with legacy networking schemes. NFV is assisting SDN by deploying virtualized network functions in the box-style hyper-converged resources of cloud data centers [21]. Virtual monitoring is used to collect box-related data for monitoring and troubleshooting purposes, which one of them is described in our previous work [22]. Moreover, the centralized orchestration for multi-tenant cloud data centers and NFV-assisted SDN infrastructure can provide the simplified orchestration of SDI-ready playground with the slice-based network virtualization support. Thus, by considering the above features, SmartX Box is designed to be ready for supporting a wide range of research experiments. Also, by merging all the required functionalities into the hyper-converged SmartX Box, it is easier to realize the scale-out capability of the playground by simply adding hyper-converged SmartX Boxes to increase the resource capacity of the playground.

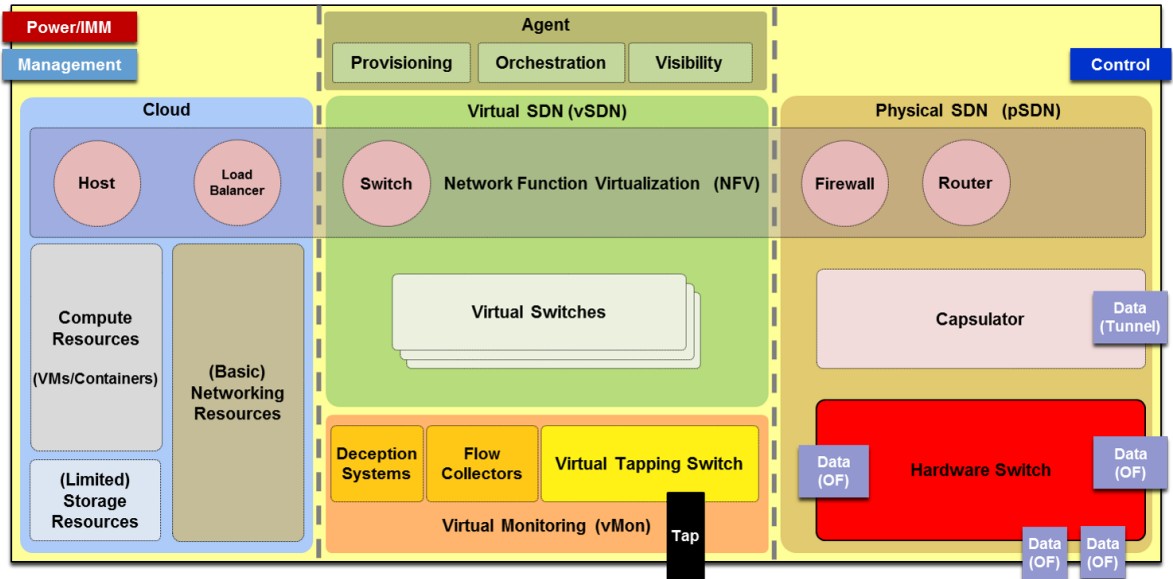

**Figure 5.** Hyper-converged SmartX Box: Abstraction to match SDN/NFV/Cloud integration.

To support the flexible remote configuration, each hyper-converged SmartX Box requires dedicated and specialized connections for P/M/C/D (power, management, control and data), which are explained in the next section [7]. However, besides those connections, there are no specific hardware requirements for hyper-converged SmartX Boxes. Therefore, any commodity hardware with reasonable computing, storage, and networking resources can be used. The total amount of hardware resources only affects the capacity (e.g., the total number of VM instances per flavor types) in specific boxes, sites, and regions. However, it is essential to consider the hardware acceleration support for virtualization and networking.

### 2.3.2. Virtualized SDN-Enabled Switches and Cloud-Leveraged VMs

As described above, the conversion from SmartX Racks to SmartX Boxes are completed to manage better the distributed multi-site cloud-based services on the top of SDN-enabled inter-connect capabilities [7]. Thus, the actual design of SmartX Box needs to consider and balance both cloud and SDN aspects carefully. The OpenStack [10] Cloud can provide VM instances and basic networking options for diverse tenants. For SDN, several instances of virtual switches (based on Open vSwitch [9]) are provisioned while allowing users/developers to share them simultaneously. We arrange the SDN and cloud relevant functions inside a single hyper-converged SmartX Box, as shown in Figure 6. SDN-related virtual functions consist of several virtual switches with different roles, e.g., creating developers networking topology, inter-connecting OpenFlow-based overlay networking, and tapping flows for troubleshooting. Also, cloud-related functions are placed to include VM instances for cloud-based applications and to support external connections to VMs.

The inside view of SDN-/cloud-related functions is depicted in Figure 7. First, several SDN-enabled virtual switches are placed and matched with its functionalities: *brcap* for capsulator (encapsulate OpenFlow packets through an overlay tunnel), *br1* and *br2* for users/developers switches, and *brtap* for tapping purpose (capturing packets for troubleshooting as described in [11]). Cloud-related VM instances (a.k.a., virtual Boxes: vBoxes) are managed by KVM hypervisors, which is controlled by OpenStack Nova with specific flavors and images. Additionally, virtual switches (i.e., *br-int*, *br-ex*, and *br-vlan*) and user-space virtual router are configured by OpenStack Neutron to provide required connectivity to cloud VM instances.

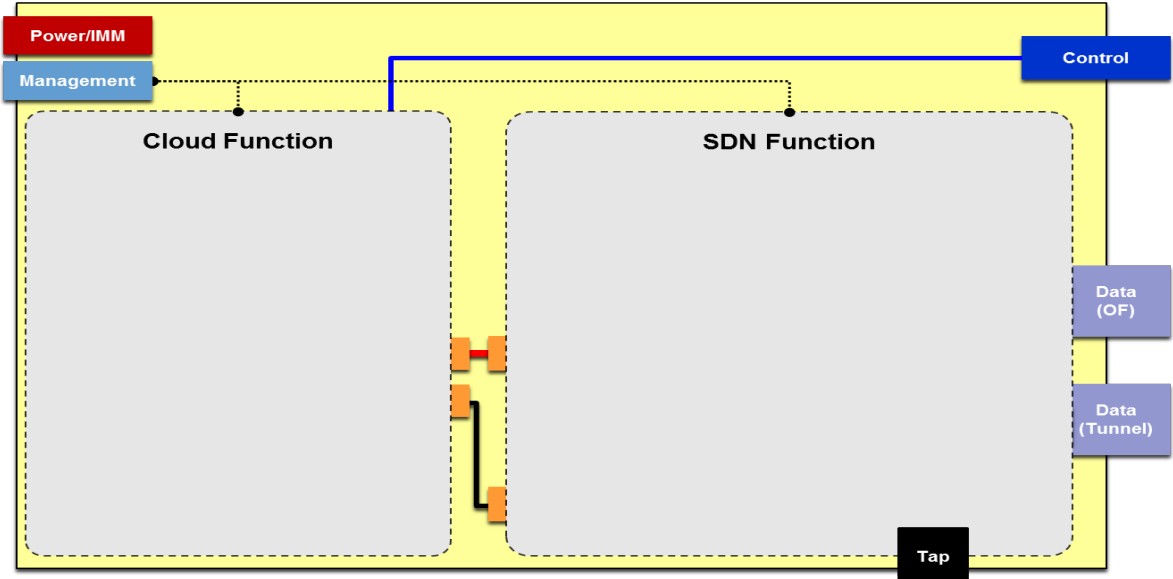

**Figure 6.** Hyper-converged SmartX Box Design for SDN and Cloud Integration.

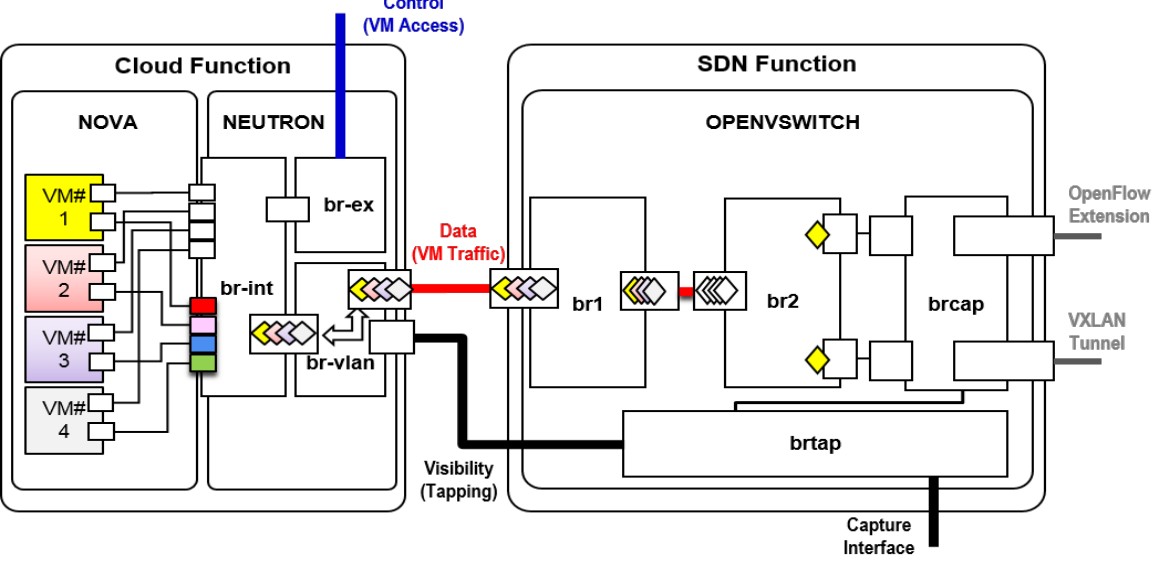

**Figure 7.** Hyper-converged SmartX Box Virtualized Cloud and SDN Components.

*2.4. Semi-automated Resource Provisioning*

Deploying hyper-converged SmartX Boxes in heterogeneous physical (i.e., network topology) infrastructures is very troublesome since it is subject to different performance parameters and independent network administrative domains. As mentioned in the previous section, it is tough to keep the sustainable operation of all the boxes. Thus, a set of automated provisioning tools is developed to minimize the consumed time for provisioning (i.e., installing and configuring) all the SmartX Boxes with pre-arranged P/M/C/D connections. The P (Power) connection is used for power up/down SmartX Box. The M (Management) connection is mainly used for managing SmartX Box by the operator. Also, the C (Control) connection is used to access and control the SDN-/Cloud-related functions (i.e., virtual switches and VMs) by the developer. Finally, the D (Data) connection is used for any data-plane traffic that includes inter-connection traffic among multiple SmartX Boxes. Also, the automated provisioning tools are controlled by a centralized P-Center inside the Playground Tower, which has full access to all distributed hyper-converged SmartX Boxes.

First, to automate the provisioning of the SDN-enabled virtual switches, *ovs-vsctl* high-level management interface for OpenvSwitch is used. Please note that *ovsdb* (OpenvSwitch database) protocol is also used for the centralized configuration of the OpenvSwitch database inside each SmartX Box. The provisioning task includes the creation of virtual switches, the configuration of virtual ports/links and overlay tunnel inter-connections, and the control connection of virtual switches and SDN controllers. Next, open-source OpenStack cloud software has special installation and configuration tools, called *DevStack*, which can support several modes of OpenStack configurations with selected operating systems (e.g., Ubuntu, Redhat Enterprise Linux, and CentOS) [23]. For OF@TEIN Playground, we customize DevStack provisioning template to facilitate multi-regional OpenStack cloud deployment with centralized management and authentication.

The overall implementation of semi-automated provisioning for SDN/cloud-enabled SmartX Box is depicted in Figure 8. It is started with a clean-up of previous software installation and checking/upgrading the operating system. Then it is followed with box installation to install/configure OpenStack cloud and OVS components, and finally API-based tools to verify function installation/configuration.

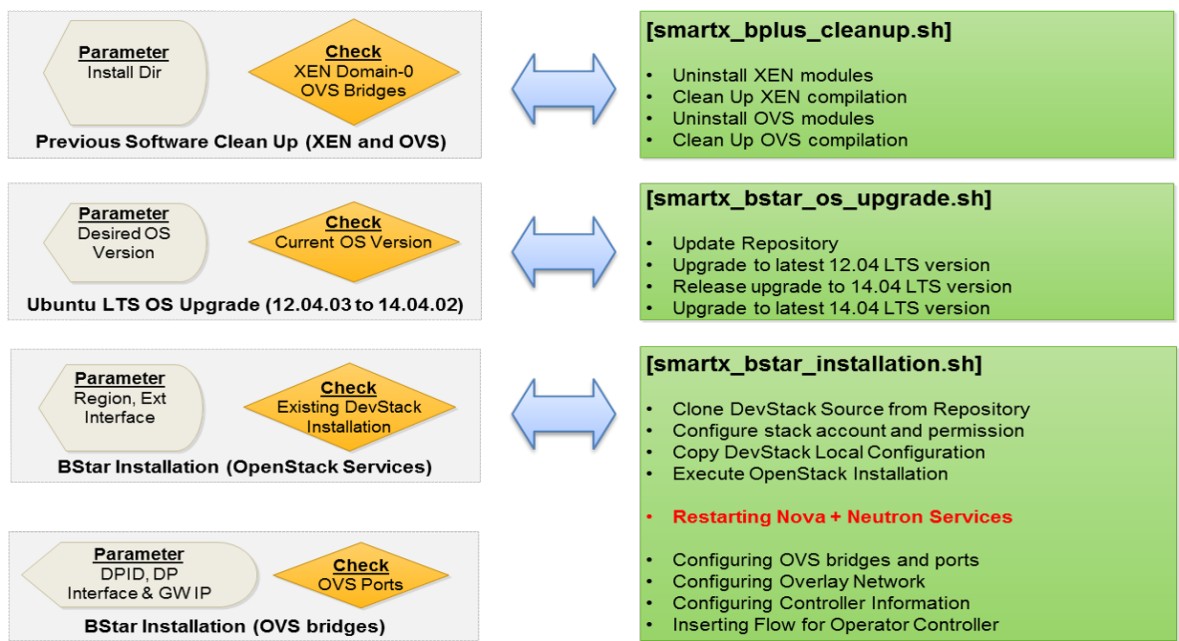

**Figure 8.** Implementation of Semi-automated Provisioning for SDN/cloud-enabled Hyper-converged SmartX Box.

*2.5. SDN and Cloud Centralized Control*

Cloud-related virtual functions are inter-connected through SDN-related virtual functions to provide end-to-end communication for multi-site cloud-based applications. The cloud-related virtual functions are controlled centrally by open-source *OpenStack* Cloud management and orchestration software [10]. The SDN-related virtual functions are also centrally controlled by open-source SDN controller such as ODL (Open Daylight) [24] and ONOS (Open Network Operating System) SDN Controller [25]. OpenStack *Keystone* provides centralized user authentication and authorization. OpenStack *Nova* and OpenStack *Neutron* can create VM instances and to provide enhanced connectivity, respectively. The SDN controller (i.e., ODL) manipulates the flow table entries of SDN-enabled virtual switches to enable the flexible steering of inter-connection flows among various functions located in different cloud sites. Both cloud and SDN control software are required to mix and match the configurations so that we can ensure the consistent connections between cloud VM instances. Remember that the main challenge is how to accommodate cloud-based multi-tenancy virtual networks (e.g., flat, VLAN, or tunneled network) for OpenFlow-based network slicing (e.g., IP subnets, VLAN IDs, and TCP/UDP ports). Eventually, VLAN-based multi-tenancy traffic

control (e.g., tagging, steering, and mapping) is chosen to integrate tenant-based and sliced-based networking in SDN-enabled multi-site clouds playground.

By manipulating both OpenStack and ODL SDN controller, a VLAN-based multi-tenancy traffic control is implemented as follows. First, we place VMs in two cloud regions and prepare the connectivity for these VMs. These VMs are tagged by OpenStack Nova with a specific tag ID. Second, OpenStack Neutron automatically maps the tag into VLAN ID that matched with SDN-based slice parameters. This matching allows inter-connection flows for VMs to be steered by the developer's SDN controller, supervised by *FlowVisor* [8]. The SDN-based flow steering inserts flow table entries according to the particular incoming and outgoing ports in the developer's virtual switches, where several ports are mapped to other cloud regions/sites. Finally, based on the destination site, it maps to a specific tunnel interface that is pre-configured by the SDN controller of operators. The example control of SDN and cloud components inside the SmartX Box for connecting several VMs from different tenants across multiple sites, as depicted in Figure 9.

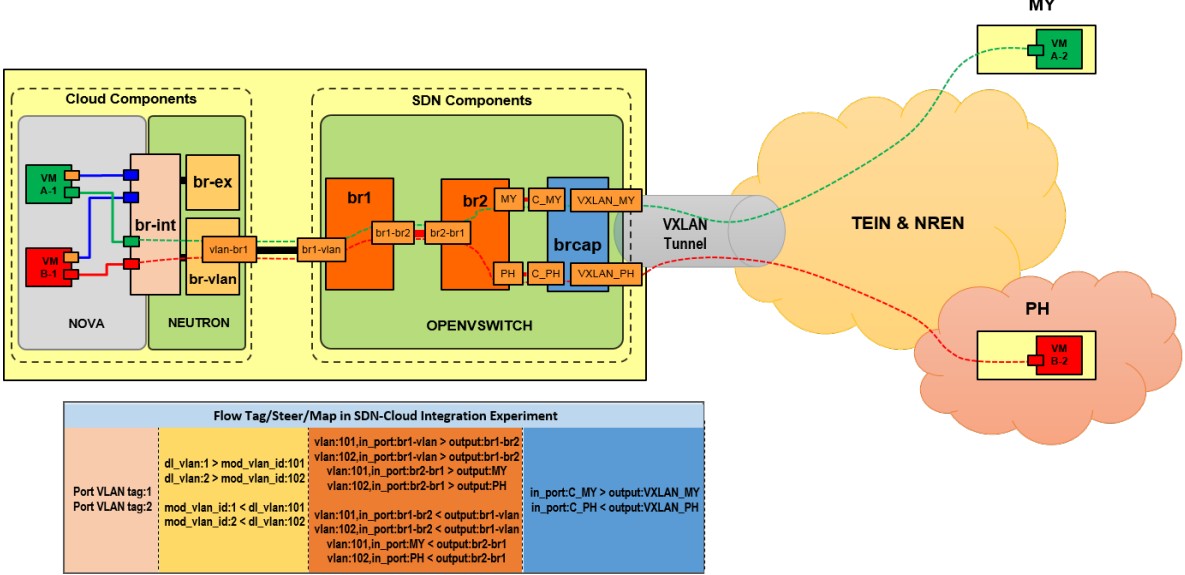

**Figure 9.** SDN and Cloud Control (Tag, Steer and Map) inside the SmartX Box.

## 3. Cost and Efficiency Analysis

### 3.1. TCO Analysis of Conversion from SmartX Rack into SmartX Box

To estimate the reduction of TCO (Total Cost of Ownership) from SDI based on the hyper-converged resources, we can observe these two reports. First, typical three-year server TCO from IDC (International Data Corporation) [26] in 2017, which defined the composition of several costs such as hardware, software, staff training, outsourced cost, user productivity, and staffing (manpower), as depicted as Figure 10a. Second, the value of SDI to reduce the total TCO of 10,000 OS instances which is released by Intel [27], as depicted in Figure 10b. Where manpower efficiency improves up to 60%, software savings up to 70%, hardware reduction up to 20%, and other reduction (infrastructure and energy) decreases up to 20%.

Based on both of the analysis reports, we can estimate the total saving of our conversion effort from SmartX Rack into SmartX Box hyper-converged resources. In summary, as depicted in Figure 11a, the TCO for SDI-ready box-style hyper-converged resources is described as follows. The hardware-related project cost reduced to 5.6% due to single box type of deployment, the software cost decrease to 2.1% due to open-source software adoption, and the most important is the cost for playground operators go down till 36%. The overall TCO saving of SmartX Rack to SmartX Box conversion is around 30.3%. If the graph is normalized into 100% of the pie chart, so we can produce

the chart as depicted in Figure 11b. In conclusion, with the same amount of project budget, more cost can be allocated for staff training (e.g., more effort for the research) to increase the quality of operators/developers and also consider more compensation on the experiment downtime or the developers/researchers productivity.

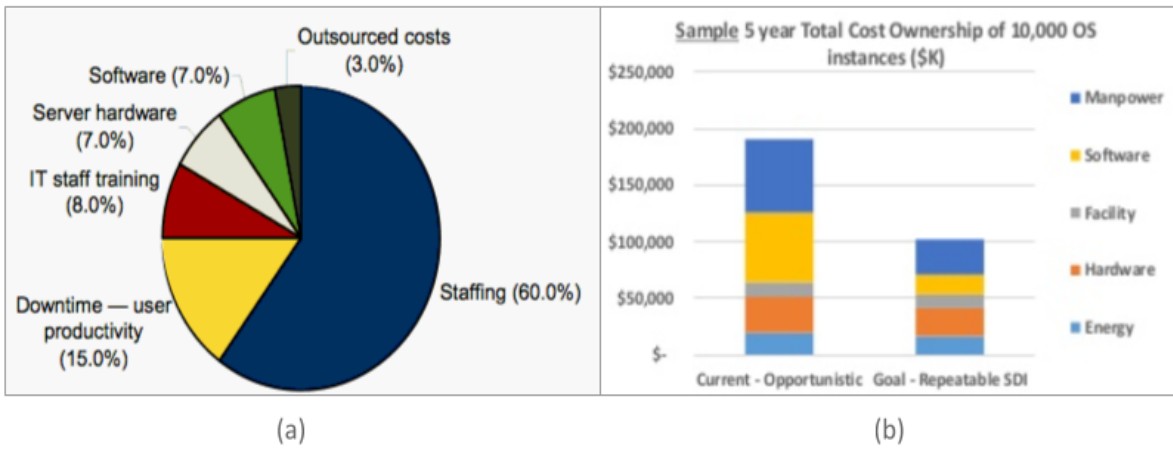

**Figure 10.** (**a**) Server-based 3-years TCO Composition [26] and (**b**) SDI Value to reduce 5 years TCO [27].

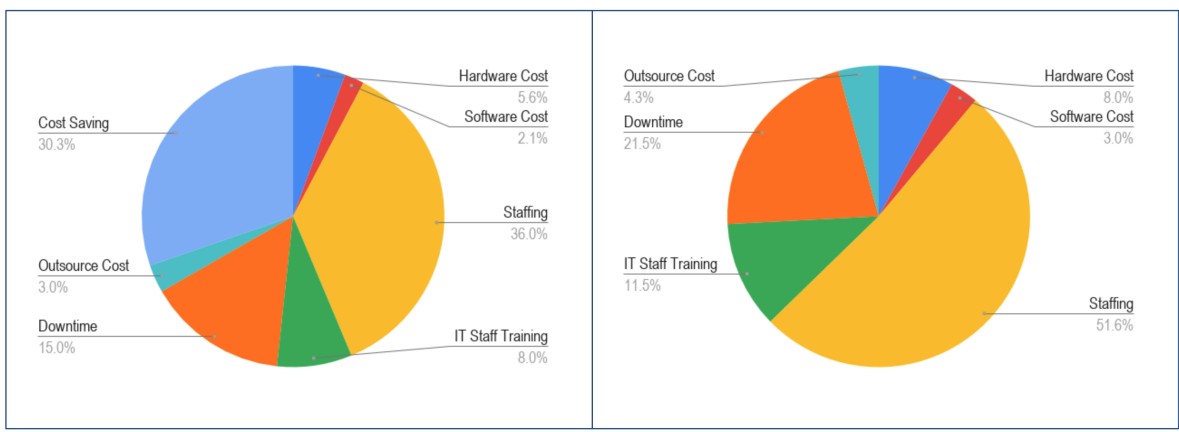

**Figure 11.** (**a**) TCO saving with hyper-converged infrastructure and (**b**) Normalized TCO with hyper-converged infrastructure.

*3.2. The Efficiency of Semi-Automated Provisioning*

To facilitate the agile deployment of OF@TEIN Playground, both SDN-/cloud-related tools are used for automated provisioning of hyper-converged SmartX Boxes. It is aligned with the recent employment of DevOps automation since the OF@TEIN Playground is operated by a limited number of operators and becomes easily uncontrollable as it spans across multi-domain inter-connected networks beyond the privileges of playground operators. Thus, by using DevStack-based OpenStack deployment and ovs-vsctl or ovsdb protocol for virtual switch provisioning, we can simplify the semi-automated provisioning of hyper-converged SmartX Boxes. Also, REST APIs of the ODL SDN controller is used for automated flow insertion, flow modification, and flow deletion. In summary, most of the provisioning steps are automated, except for manual handling of critical tasks such as DevStack-based OpenStack service restart and VXLAN tunnel checking/recovery.

The duration of the whole process of semi-automated provisioning depends on the Internet connection speed due to the online OS upgrade and online OpenStack software installation. However, it takes less time for new box clean installation (including box restart), because previous software clean-up and OS upgrade is not required. Moreover, re-configuring pre-installed SmartX Box is much faster with "offline mode" enabled because online software/package copy from Ubuntu and OpenStack repositories are not required. Figure 12a shows the semi-automated provisioning results, which take approximately 50 min for fully upgrading a SmartX Box with network connection up to 300 Mbps. Thus, it takes around 6 h for the slowest network connection which less than 10 Mbps. However, it takes only around 20 min (including box restart) for provisioning without cleaning up the previous installation and upgrading the operating system. While Figure 12b justifies the longest completion time is for upgrading the operating systems and restarting the SmartX Box. The clean-up task is negligible because it is less than one minute, and then the installation task is reasonable for such a customized configuration. Moreover, it takes less than 10 min to recover or re-configure SmartX Box with "offline mode" [7].

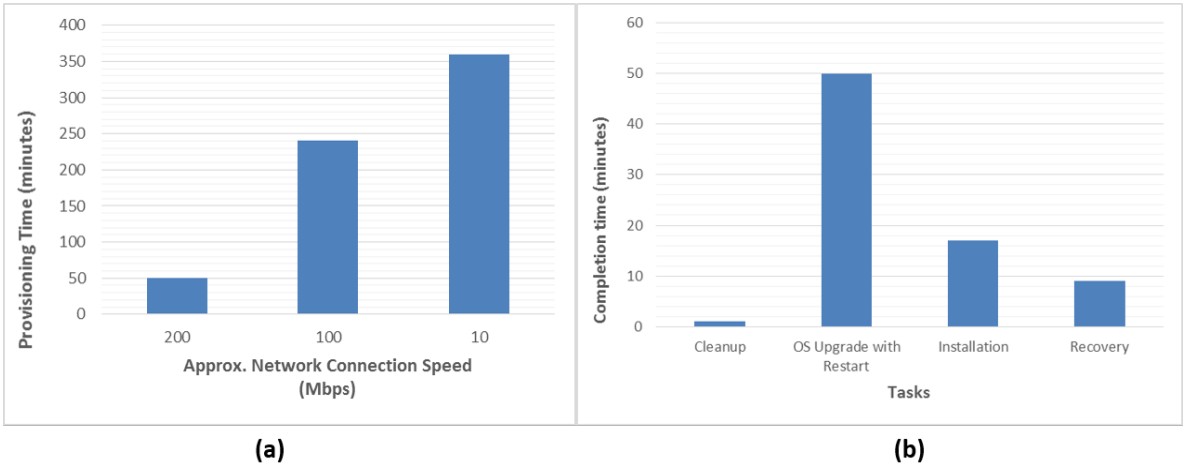

**Figure 12.** Semi-automated Remote Provisioning Result for SmartX Box: (**a**) Provisioning time for the different network connection speed, and (**b**) Completion time for different provisioning task.

## 4. Playground Deployment Verification

### 4.1. Distributed Deployment of SmartX Boxes for Building OF@TEIN Multi-Site Playground

Multiple hyper-converged SmartX Boxes are deployed on existing hardware of OF@TEIN Playground with a special focus on adding the open-source OpenStack cloud-management software. The OF@TEIN Playground relies on the heterogeneous physical underlay infrastructure across multiple administrative domains. Thus, the multi-regional OpenStack cloud deployment is currently investigated as the deployment option because it gives simple and common configuration for all regions (i.e., SmartX Box sites). It also supports an independent IP addressing scheme and has less dependency on the overlay networking among regions. Despite multi-regional independent cloud deployment, the OF@TEIN Playground supports an integrated cloud management interface by deploying web-based OpenStack Horizon UI and the centralized account/token authentication from OpenStack Keystone. The resulting OpenStack multi-regional cloud deployment is illustrated in Figure 13.

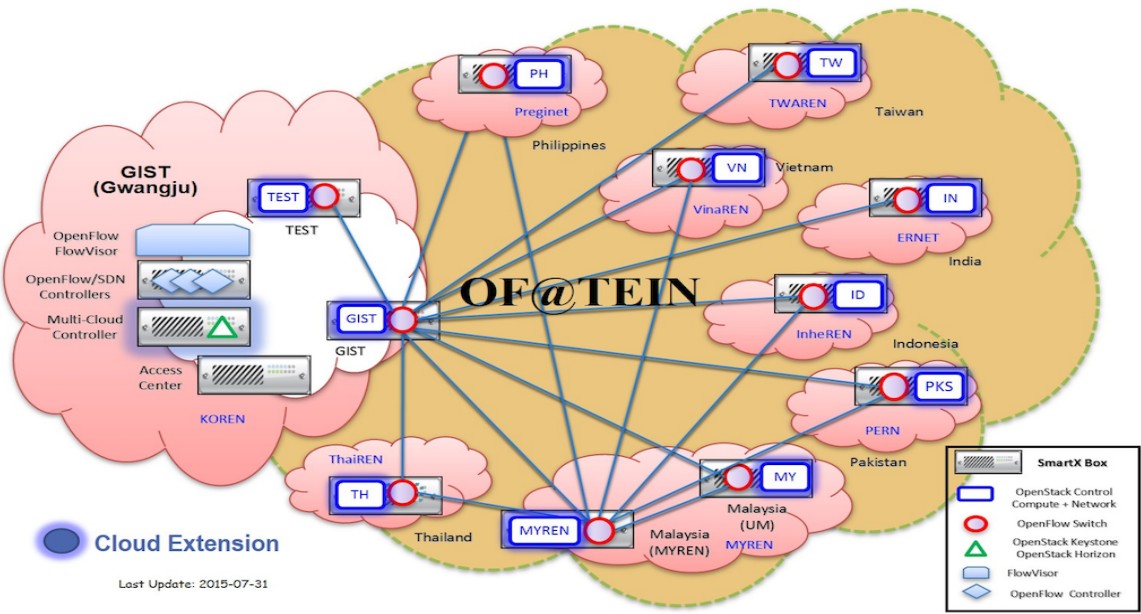

**Figure 13.** OpenStack Multi-Regional Configuration in OF@TEIN Playground.

Next, the OF@TEIN Playground is enhanced with multiple mesh-style inter-connections of NVGRE/VXLAN overlay tunnels, along with a unique flow-tapping virtual switch [22]. The OpenStack multi-region deployment is modified to build an SDN-enabled multi-site playground where inter-VM connectivities between cloud VMs are used by leveraging OpenFlow-enabled data planes. The data planes are programmed and controlled by the centralized SDN controller, co-located with centralized cloud management.

*4.2. Example of Experiment with SDN-Enabled Multi-Site Cloud Playground*

This example shows both aspects of experiment preparation in SDN-enabled multi-site clouds playground to provision resources in both OpenStack and ODL SDN controller. First, a VM in one of the cloud regions (i.e., playground sites) is prepared, including the basic connectivity for this VM using the first VM virtual NIC (Network Interface Card). It is connected to a control network, called "private", for providing VM remote access from an external network. Then, second VM virtual NIC is connected to a data network, called "datapath01", which is automatically mapped into pre-configured VLAN ID that matched with SDN-based slice parameters. This mapping allows the flow from this VM to be steered by the developer's SDN controller, supervised by FlowVisor [8]. Another VM in the second region also is prepared with the same steps as depicted in Figure 14. Next, the SDN-based flow steering that is leveraging the ODL SDN controller inserts flow table entries in developers' virtual switches from/to those VMs to/from pre-defined ports that are already mapped to other cloud regions/sites. Those pre-defined ports are mapped to a tunnel interface from the originating site into the designating site that is controlled by the operators' SDN controller of operators. The example of the steps is depicted in Figure 15.

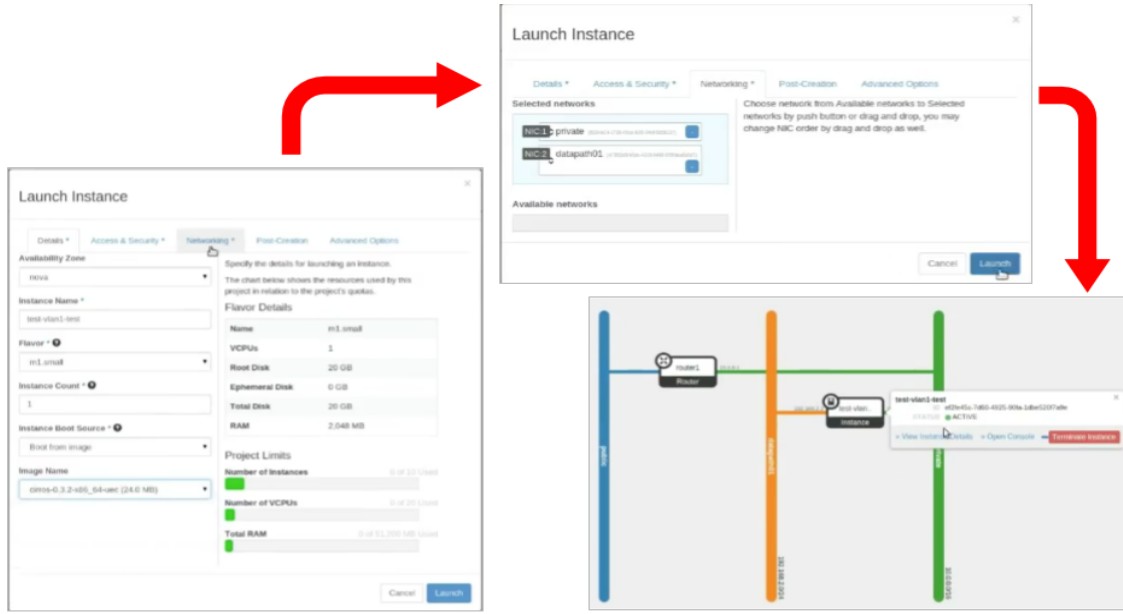

**Figure 14.** OpenStack VMs Preparation and Configuration.

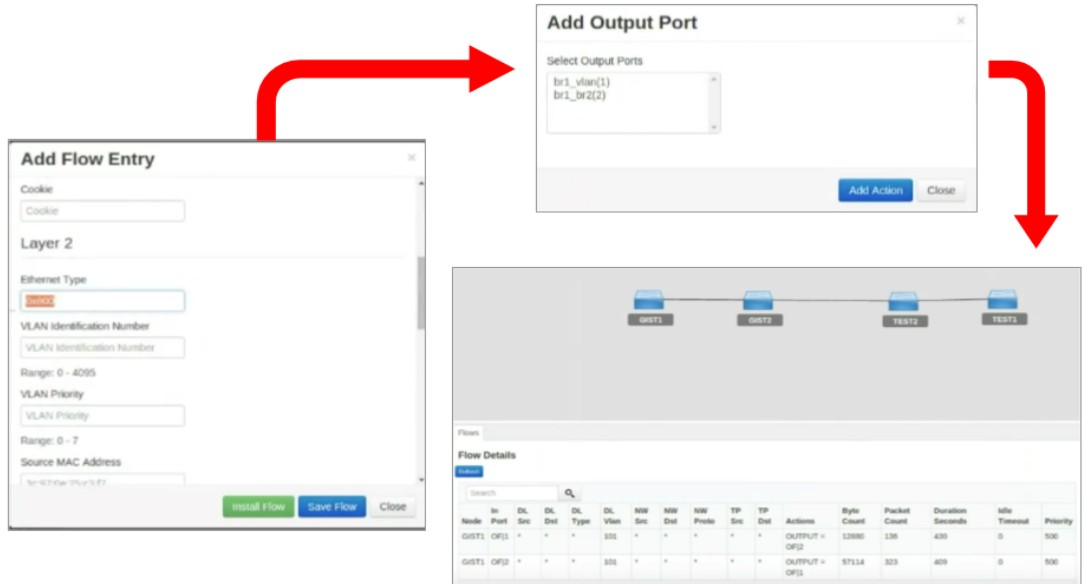

**Figure 15.** Flow Configuration in OpenDaylight Controller.

*4.3. Multi-site Playground Visibility and Visualization*

Resource-level monitoring and visualization are important operation activities for OF@TEIN playground. The SDN and cloud components implementation for hyper-converged SmartX Boxes provides diverse physical and virtualized resource combinations while at the same time brings new complexities for monitoring and visualization. The SDN-cloud-enabled playground demands considerably different resource-level visibility solutions from traditional networking testbed. A distinctive, component-based, data-oriented approach is required for resource-level visibility of distributed OF@TEIN physical and virtual resources. By integrating open-source software/tools, we set up a unique resource-level visibility solution, which is focused on operation data collection from multiple sources and interactive large-scale visualization [28]. Resource-level visibility data is collected in nearly real-time to help the operators for monitoring the status of the resources by using a single and unified visibility user interface. Large-scale visualization (i.e., network tiled display leverages SAGE Framework [29]) enables the simultaneous visualization of the multiple and different types

of visualization sources (e.g., web-based UI, remote desktop, secure shell), as depicted in Figure 16. It allows OF@TEIN operators to manage the resources and developers to execute the experiment while keeping an eye on the playground resources.

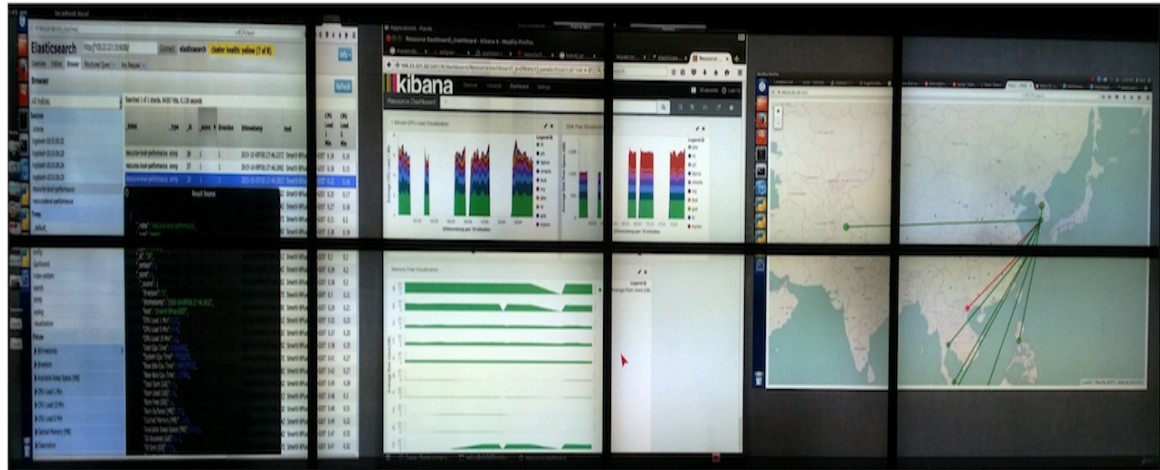

**Figure 16.** Multi-site playground resources visibility over network tiled display.

## 5. Conclusions

This paper gives comprehensive discussions of the unique concept and design of SDN-enabled multi-site clouds playground with hyper-converged box-style resources for an innovative and diverse research experiment. OF@TEIN Playground is successfully provisioned as an affordable SDN-enabled multi-site clouds playground with distributed SmartX Boxes deployment for integrated SDN and cloud experiments. We believe that an open, agile, and economics box-style hyper-converged resources can provide a larger scale of an affordable and sustainable playground for diverse experiments with a wide variety of application areas.

**Author Contributions:** Conceptualization—J.K. and A.C.R., Supervision—J.K., Investigation—A.C.R., Software and Validation—A.C.R., and M.U., Visualization—M.U., Writing—A.C.R., and Reviewing and Editing—J.K. and M.U.

**Funding:** This work was supported by Institute of Information & Communications Technology Planning & Evaluation (IITP) grant of the Korea Government (MSIT) (No. 2015-0-00575, Global SDN/NFV Open-Source Software Core Module/Function Development, and No. 2017-0-00421, Cyber Security Defense Cycle Mechanism for New Security Threats). This research is also partially supported by Asi@Connect grant of the Asi@Connect-17-094 (No. IF050-2017), OF@TEIN+: Open/Federated Playground for Future Networks.

**Acknowledgments:** The authors would like to acknowledge gratefully to OF@TEIN Community who has been supported the deployment and operation of OF@TEIN playground for the past few years. Especially for playground operators and NREN administrators who help us to maintain the stability and connectivity of the resources to be able continuously used by playground developers/users. Hopefully, the collaboration amongst all the involved parties can be continued to enhance and extend our proposed concept and effort in the next few years.

**Conflicts of Interest:** The authors declare no conflict of interest.

## Abbreviations

The following abbreviations are used in this manuscript:

| | |
|---|---|
| SDN | Software-defined Networking |
| GENI | Global Environment for Network Innovations |
| FIRE | Future Internet Research and Experimentation |
| OF@TEIN | OpenFlow at Trans-Eurasia Information Network |
| DevOps | Development and Operations |
| LAN | Local Area Network |
| VM | Virtual Machine |

| | |
|---|---|
| CNG | Cloud Networking Gateway |
| SDI | Software-defined Infrastructure |
| KVM | Kernel Virtual Machine |
| LXC | Linux Container |
| OVS | OpenvSwitch |
| NF | Network Function |
| SD-WAN | Software-defined Wide-area Network |
| IoT | Internet of Things |
| MaaS | Machine as a Service |
| API | Application Programming Interface |
| CLI | Command Line Interface |
| NFV | Network Function Virtualization |
| IPMI | Intelligent Platform Management Systems |
| ODL | Open Daylight |
| ONOS | Open Networking Operating System |
| VLAN | Virtual LAN |
| TCP | Transport Control Protocol |
| UDP | User Datagram Protocol |
| TCO | Total Cost Ownership |
| IDC | International Data Corporation |
| NVGRE | Network Virtualization using Generic Routing Encapsulation |
| VXLAN | Virtual Extensible LAN |
| REST | Representational State Transfer |
| SAGE | Scalable Adaptive Graphics Environment |

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
