# Peer review of "SmartX Box: Virtualized Hyper-Converged Resources for Building an Affordable Playground"

_electronics, doi:10.3390/electronics8111242_

Round 1
Reviewer 1 Report
This paper presents how to build an affordable playground for SDN-Cloud experiments by
utilizing hyper-converged distributed SmartX Boxes. This paper has some merit and can be
published, provided that the authors address all of the topics listed below.
1. In the Introduction section, the authors refer to the SmartX Rack configuration. I think it is of great
importance to describe the type of network being used (network interconnections, topology etc).
Special focus should be given to the OpenFlow switch interconnections, as the
collection of OpenFlow-enabled virtual switches is a key factor for the implementation described.
Two works that describe such typical interconncections for OpenFlow switches that should be referred to are:
1a) Feilong Tang, Laurence T. Yang, Can Tang, Jie Li Senior, and Minyi Guo A Dynamical and Load-Balanced
Flow Scheduling Approach for Big Data Centers in Clouds, IEEE Transactions on Cloud Computing,
2018,6, pp. 915–928.
1b) Stavros Souravlas, "ProMo: A Probabilistic Model for Dynamic Load-Balanced Scheduling of Data Flows in Cloud Systems ",
Electronics, Open Access Journal, vol.8, issue 9, September 2019
2. In my opinion, the contributions of this work should better be described in more details after describing the
related work, and emphasis should be given to the points that this newly presented work differs/improves
from other similar works. Do not just mention the contributions of this work, but also some strong points
compared to the others. Generally, this Introduction section requires some kind of re-organization. I would prefer
to see a typical title like Introduction-Related Work with no subsections like 1.1, 1.2, 1.3 (these
subsections are rather small, there is no need for sectioning). In this section, you can introduce your work,
describe the network (as suggested in the previous comment), stress your motivation, describe the previous work,
stress the main differences of the suggested work and your contributions. I also think that the
related works (there are only 4) should be discussed in more details, to give the reader a better idea
where this work really stands.
3. This paper introduces some ideas on how to build the playground, but from the description, it is rather
unclear to the reader. For example, all of the parts presented in Fig. 4 need to be described, as an unfamiliar reader may wonder what
Virual Monitor or Virtual Tapping Switches are doing. When presenting a new hardware/software based solution,
it is absolutely necessary to fully describe it. Generally, all the figures of this section need to be
described. To minimize the effort, the authors could give a brief definition/description of each single part's
work. That would be really helpful and improve the paper's clarity and quality.
4. In the experimental section, it is unclear to me how the proposed system is verified. The title says
Verification and Measurement, but in my opinion, it should be re-organized as
Section 3: Cost Analysis and Efficiency
Section 4: Experiments
This is because Paragraphs 3.1-3.3 can be separated from the Experimental examples that follow. This would
improve the readability.
5. In Section 3, page 13, there is a typo -> The duration on the whole process of
semi-automated provisioning is depends...... please remove the word is.
Also, on the same page, However, it takes less duration should be replaced by -> it takes less time
Generally, work more on some typos of this kind.
Author Response
Dear Reviewer,
Please find my response in this attachment.
Best regards,
Aris.

Reviewer 2 Report
Considering the title I would recommend to be adapted to: SmartX Box: Virtualized Hyper-converged Resources for Building an Affordable Playground.
The authors provided an affordable distributed SDN-Cloud testbed by utilizing hyper-converged SmartX Boxes that are distributed across multiple sites, integrating both cloud multi-tenancy and SDN-based slicing, aiming to facilitate resources management, and allow developers to run experiments.
The combination for distributed SDN and Cloud testbed aims at providing scalable and flexible computing resources with enhanced networking capability. The resulting playground is sought to be composable (e.g., programmable) in terms of computing, storage, and networking types of resources, integrated into a single box-style entity, to enable customization and scalability without the manual intervention of playground operators.
The authors claimed that they use hyper-converged SmartX Boxes, which had already been designed to virtualize and merge the functionalities of four devices into a single box, and thus provide a more compact solution. However, in the introduction, the relation between SmartX Racks and SmartX Boxes is not clear and it should be clarified properly, along with the functionalities of the SmartX Boxes (which is partially provided in lines 120-125). Also, in related work the authors should mention the previous applications of SmartX Boxes that exist in the literature.
The seamless integration of SDN-enabled and cloud-leveraged infrastructure is claimed by the authors to be a very challenging task, due to open and conflicting options in configuring and customizing resource pools together. The authors try to address the challenges described in lines 201-211.
In lines 157-160 the authors mention that “in our approach, Playground Tower provides a logical space-like abstraction in a centralized location, which leads the operation of the multi-site playground by following the concept of “monitor & control” tower. From the tower, the DevOps team can enjoy a panoramic view of playground resources that are distributed over underlay networks and quickly respond to operational issues.” However, it should be made clear if the “Playground Tower” is an innovative architectural component proposed by the specific paper or if it had been developed before and in which concepts. This applies to each architectural component that is being presented (e.g. Virtualized SDN-enabled Switches and Cloud-leveraged VMs), to illustrate the originality/novelty of the paper.
Line 205 includes a reference that is not readable.
In Figure 9, in the caption, it should be mentioned the source of the figures.
Considering the Verification and Measurement section, it is not straightforward how the TCO saving with hyper-converged infrastructure is computed. Also, in lines 331-332, it is mentioned that “Multiple hyper-converged SmartX Boxes are deployed on existing hardware of OF@TEIN Playground with special focus on adding the open-source OpenStack cloud management software”. The resulting contribution should be clarified.
Section 5, “Discussion”, should rather be placed in ‘Related Work”, adding a sub-section “Aim of the Paper”.
Author Response
Dear Reviewer,
Please find my responses in this attachment.
Best regards,
Aris.

Round 2
Reviewer 1 Report
I believe the paper is now improved and can be considered for publication.
Author Response
Thank you very much for your reviews.

Reviewer 2 Report
Overall, the paper is improved. Some comments considering minor issues follow:
lines 27-28: "The comparison between previously deployed SmartX Racks and newly deployed SmartX Box, as depicted in Fig. 1." Should be: "The comparison between previously deployed SmartX Racks and newly deployed SmartX Box, is depicted in Fig. 1."
line 90: Should be: "They are a little bit concerned..."
lines 93-95 should have a better structure.
line 107: Should be: "To centrally manage..." Also in this sentence mention that you utilize a Playground Tower, as a centralized controller.
Author Response
Thank you for your reviews.
Please find attached my responses.
